# Multiscale Elastic Modulus Characterization of Ti_0.5_Fe_0.45_Mn_0.05_, an Iron–Titanium–Manganese Alloy Dedicated to Hydrogen Storage

**DOI:** 10.3390/ma17246100

**Published:** 2024-12-13

**Authors:** Ludovic Bebon, Anne Maynadier, Yves Gaillard, David Chapelle

**Affiliations:** 1Department of Applied Mechanics, FEMTO-ST Institute, CNRS, Université de Franche-Comté, 25000 Besançon, Franceanne.maynadier@univ-fcomte.fr (A.M.);; 2Fuell Cell LAB––FCLAB (UAR2200), 90000 Belfort, France

**Keywords:** hydrogen storage, TiFe alloy, intermetallic, elasticity, mechanical characterization

## Abstract

Hydrogen storage in intermetallic compounds, known as solid-state storage, relies on a phase change by the metal alloy. This phenomenon causes a violent change in volume at the crystalline scale, inducing a change of volume for the millimetric particles and, with time, important stresses on the tanks. It is thus necessary to know the mechanical behavior of the material to report these phenomena and improve the tanks’ reliability. The present study deals with the mechanical characterization of Ti_0.5_Fe_0.45_Mn_0.05_ alloy at different scales. First, the elastic modulus was measured by compression tests of cylindrical samples. The estimated macroscopic elastic modulus was about 198 GPa, with high variability, from 163 to 229 GPa. Secondly, ultrasonic elastic characterization together with instrumented indentation allowed an estimation of both Young’s modulus and Poisson’s ratio at 269 GPa and 0.29, respectively. Finally, the nanoindentation results, combined with SEM imaging and EDS analyses, revealed that several metallurgical phases coexist below the particle scale. Four distinct domains in terms of elasticity were clearly identified. The coherence of all these estimations is discussed and interpreted considering the true microstructure of the material and the defects present in the different samples.

## 1. Introduction

Hydrogen is considered a good candidate to serve as an energy carrier in the context of the energy transition in the face of the urgency of climate change. Safe and energy efficient storage solutions are a prerequisite for the green nature of the hydrogen supply chain. The production of clean energy sources (solar, wind and hydro) is intermittent and rarely coincides with energy consumption needs. It is therefore mandatory to set up storage facilities. Solid-state hydrogen storage has the advantage of operating at pressures and temperatures close to ambient (between 1 and 50 bars and 0 and 80 °C), unlike storage by gas compression (typically between 200 and 700 bars), and unlike storage in the liquid state (at temperatures below −252 °C). This makes it a safer, more sensible and more socially acceptable option.

The reversible hydriding on which solid-state hydrogen storage is based is associated with a violent change in volume at the crystal scale: up to 30% [1]. The storage material, generally crushed into powder, undergoes repeated swelling and shrinking, often described as breathing. Although the causes have not been formally identified, self-fragmentation and particle decrepitation have been reported for years in many storage alloys ([2] for TiMn1.5, [3] for TiFe and [4] for LaNi5). Joubert et al. [5] and Zeaiter et al. [6] evaluate the morphological changes and the refinement of powder bed granulometry for industrially used LaNi5 or TiFe, for example.

At the reservoir scale, these phenomena cause functional problems of compaction or damage to the container. Okumura et al. [7] imaged the decrepitation occurring over in situ sorption–desorption cycles using X-ray Tomography and reported a decrease in void volume and the occurrence of packing in the lower region of the vessel. Charlas et al. [8], by measuring the displacement of the free surface of a Ti-V-Cr alloy powder bed at a small storage cell scale, experimentally highlighted the swelling and shrinking of the bed during the hydrogen absorption cycles. They also showed that the resulting progressive compaction depends on the external pressure applied to the sample on the upper free surface but is also due to friction on the wall. Large mechanical stresses and strain on the container walls have been regularly reported, measured either using strain gages [9,10,11] or digital image correlation [12]. They are particularly intense in the very first cycles and mainly located on the bottom of vertical vessel, where fine particles migrate and the bed compacts. This jeopardizes the efficiency and security of this storage technology, sometimes leading to rupture [13]. Thus, this problem appears to be of major interest for the industrial implementation of solid-state storage.

In the past decade, modeling approaches were also used. Among them, Bebon et al. [14] explored the causes of the particle size stabilization after a certain number of hydriding cycles in an analytical mechanical model. The Discrete Element Method [15] has been identified as relevant by several research groups to model either the rupture of a single particle or the behavior of the powder bed (breathing, compaction, etc.) [16,17,18]. All these numerical modeling studies require the knowledge of the mechanical behavior of the considered storage alloy, at the scale of the massive material or at that of the particle. It shows the necessity of this study which concerns the mechanical characterization of a TiFe-based alloy dedicated to hydrogen storage at different scales.

The studied material is a titanium–iron alloy with substitution of iron by manganese: Ti_0.5_Fe_0.45_Mn_0.05_. The TiFe alloy is the main representative of the AB family of solid hydrogen storage materials. The physical and chemical properties have been widely studied, and the mechanical properties of TiFe have been derived from the crystalline structure by Benyelloul et al. [19]. Ti_0.5_Fe_0.45_Mn_0.05_ was chosen as the study material because it is a good intermetallic storage compound, as evidenced by its high storage capacity (1.68 wt% at 25 °C), good sorption kinetics, reversibility (almost full) and ability to work in mild temperature and pressure conditions (between 0 and 50 °C, between 1 and 40 bar), and is already successfully used in industrial solid storage tanks. It also has the advantage of being composed mostly of the elements Fe and Ti, which are easily sourced and recycled.

The addition of a small amount of Mn reduces the few defects [20]: the substitution of Fe by Mn minimizes its sensitivity to pollutants and therefore facilitates its activation; increases the diffusivity of H by enlarging the cell volume compared to TiFe and favors micro-cracking, resulting in improved absorption–desorption kinetics [21,22]. For more information on its behavior in response to hydrogen, the reader is invited to consult the recent reviews [23,24].

It should be noted that almost all recent studies on TiFe-based systems and related alloys (addition of -Mn, -Ni, -Co or -V, for example) are carried out from a chemical or metallurgical point of view and that few publications have been interested in its mechanical behavior, which is critical for its real implementation in technical storage solutions. This article presents in its first section the considered TiFeMn alloy and its elaboration process. The preparation of samples giving access to centimetric to nanometric scales is presented, and then the different experimental methods used to evaluate the elastic modulus are detailed: longitudinal ultrasonic wave propagation, compression tests instrumented by digital image correlation and nanoindentation combined with analyses by energy-dispersive X-ray spectroscopy and backscattering SEM imaging. The second part is devoted to estimating the elastic modulus according to the three different mechanical approaches. In particular, it is highlighted that each experimental technique loads the sample at different scales, leading to different elastic modulus estimation. Finally, in a third part, based on the ternary phase diagram of TiFeMn and on the theorical elastic responses of TiFe alloys, an interpretation of the elasticity of the different phases composing the samples is given.

## 2. Materials and Methods

### 2.1. Sample Preparation

An ingot was produced by MAHYTEC Company (Dole, France) using an induction furnace, under a controlled atmosphere (argon), by melting at 1300 °C the supposedly pure raw components (purity greater than 99.3%) into a zirconium dioxide crucible to be cast in a rectangular copper mold. The alloy is then cooled in the furnace, still under argon gas. Usually, the ingot is then crushed to make it easy to fill the tank and to first activate the hydride. Hereinafter, to perform macroscopic mechanical characterization, we deal with material specimens sampled from the as-cast ingot.

Cylinder samples with a nominal diameter D = 10 mm and a nominal height h = 15 mm were withdrawn from the ingot by electrical discharge machining (EDM), and the circular surfaces were polished. The sampled cylinders showing emerging internal cracks were immediately excluded from testing. Eight testable cylindrical samples were obtained, with a discrepancy of ±0.2 mm in diameter and ±1 mm in height. One was used to verify the experimental settings. All the cylindrical samples were subjected first to ultrasonic characterization, then to compression tests.

In parallel, a sample of a few cubic centimeters from the same ingot was polished. Surface preparation was performed using polymeric discs covered with diamond particles from 30 µm to 0.1 µm. Final polishing was realized with colloidal silica without any thermal treatment or chemical attack in preparation for micrographs and nanoindentation tests.

### 2.2. Experimental Methods

#### 2.2.1. Ultrasonic Elastic Characterization

The elastic modulus can be identified thanks to the velocity of elastic waves in the material. Characterization with elastic waves is fast, gives a macroscopic result and is nondestructive assuming an isotropic and homogeneous material. We used an ultrasonic wave transmitter/receiver (integrated Sofranel device, Sartrouville, France) to propagate elastic waves along the longitudinal direction of the cylindrical samples. Successive echoes were recorded and displayed with an oscilloscope. The wave velocity was inferred from the reflected waves’ time delay and the distance between the parallel faces of the samples (≈15 mm).

For a homogeneous material, the elastic modulus E is related to the wave velocity by [25]:(1)$$E=A\rho V^{2},$$
with ρ being the density of the sample; V the longitudinal velocity of the waves; and A a material constant, related to the Poisson’s ratio ν through the following expression:(2)$$A=\frac{\left(1-2\nu \right)\left(1+\nu \right)}{1-\nu }.$$

#### 2.2.2. Compression Test

The traction test is the classical experimental method to assess the elastic modulus. Given the difficulties of machining, due to material fragility, and the size of the possible sample, compression tests have been chosen here. Compression tests are of particular interest to obtain a macroscopic measure, with the possibility of verifying the loading and response at the local scale, with digital image correlation. Compression tests were carried out with a 100 kN Instron electromechanical universal testing machine with two compression plates. For each sample, the contact surfaces were lubricated with metal grease to limit the barrel effect due to friction and keep the uniaxial compression loading. The applied compressive solicitation was controlled in displacement with a velocity of 1 mm/min. Repeated progressive loading was applied in order to measure the elastic modulus several times during a single compression test, and then compression is applied until the sample collapsed. Force was measured along the compression axis with a 100 kN MTS load cell on the crosshead. Displacements were recorded on the encoder wheels of the testing machine, but the flexibility of the whole device created a non-negligible discrepancy between the prescribed displacement and the real displacement applied to the sample. To measure displacement and derive strain on the specimen surface, Digital Image Correlation (DIC) was performed. Two flat LED lights were used to illuminate the specimen, with a slight inclination relative to the optical path. A Point Grey camera (Grasshopper3, Richmond, BC, Canada) with a resolution of 2048 × 2048 pixel^2^ and 256 gray levels was mounted. The camera was coupled with a Schneider Xenoplan 2.0/28 lens (Bad Kreuznach, Germany), and it acquired an image every 0.5 s.

The image acquisition was synchronized with the load cell force and displacement of the cross head. Figure 1 shows the first image, before complete contact. DIC used a random high-contrast pattern, called a speckle pattern, on the surface to measure displacement. To obtain such a coating, each cylinder was first painted white before being passed into a cloud of black paint. The largest black dots had a size around of 0.152 mm.

DIC was performed using the software UFreckles, Version 2.0 [26]. This global correlation software assumes a continuous field of displacement, expressed at the nodes of a finite element mesh, calculated by the global minimization of error over the whole studied area. In Figure 1, the white rectangle represents the area of analysis (AoA). Indeed, DIC assumes a flat and immutable surface with the exception of the strain to be measured; that is why the analysis area was restricted to the central third of the specimen, where the limited curvature induces a variation in depth of only 0.28 mm. Moreover, the lighting was almost uniform over the AoA. The white frame also avoided the ends of the specimen, in contact with the plates, which were gradually covered by the grease expelled under the effect of compression. DIC analysis was performed in two stages. First, a fine discretization (element size 20 × 20 pixel^2^ or 0.27 × 0.27 mm^2^) was used in order to ensure that the displacement field was uniform and correctly longitudinally oriented, as well as to detect breakage, localization or delamination of the paint, which terminated the validity of the DIC analysis. Then, a second DIC analysis was performed with larger elements (element size 40 × 40 pixel^2^ to 80 × 80 pixel^2^) in order to minimize the computation error of displacement and strain quantities [27,28]. The longitudinal strain was extracted from a central numerical strain gage (200 × 175 pixel^2^ 2.7 × 2.4 mm^2^) and compared to the engineer stress.

The engineer stress used hereafter is defined by the ratio of the force measured by the load cell to the initial section of the sample. This stress and the strain derived from UFreckles DIC made it possible to plot the true stress–strain curves for each sample from the beginning of the test until the loss of homogeneity or the appearance of cracks in the paint, explaining the different possible number of charge cycles. Assuming volume conservation during the deformation, engineer strains, $\varepsilon_{0}$, were deduced from longitudinal true strains, ε, and then true stress values, σ, were calculated from the following formulae:(3)$$\varepsilon =ln\left(1+\varepsilon_{0}\right)$$
(4)$$\sigma =\frac{F}{S_{0}}\left(1+\varepsilon_{0}\right)$$

#### 2.2.3. Micro-Scale Measurement

The polished sample was used to make micro-scale measurements. Three complementary methods were deployed on the surface of the Ti_0.5_Fe_0.45_Mn_0.05_ sample: Scanning Electron Microscopy (SEM), nanoindentation testing and energy-dispersive spectroscopy (EDS). Two squares of 50 µm size each are swept by 100 indent tests distributed over a 10 by 10 matrix, separated with 5 µm with an Anton Paar (Baden, Switzerland) ultra-nanoindenter device mounted with a Berkovich tip. The apparent indentation elastic modulus, $E_{a}$, and hardness, H, were extracted from the load–displacement curves using the method described by Oliver and Pharr [29]. The shape area function was calibrated using fused silica. The indentation elastic modulus M was calculated using the following formula:(5)$$\frac{1}{M}=\frac{1}{E_{a}}-\frac{1-\nu_{i}^{2}}{E_{i}},$$
where $E_{i}$ and $\nu_{i}$ are, respectively, the Young’s modulus and the Poisson’s ratio of the indenter, estimated at 1141 GPa and 0.07. For an isotropic material, M depends on the Young’s modulus E and the Poisson’s ratio ν of the indented material:(6)$$M=\frac{E}{1-\nu^{2}}.$$

SEM was carried out to observe the surface with a Quanta 450 W apparatus. The local composition of the alloy was determined with an Everhart–Thornley SE detector probe with 5 nm resolution and 3 min counting time.

These three techniques made it possible to measure locally the apparent elastic modulus, the hardness, their spatial distributions and the composition of the alloy. A methodology inspired by Ulm et al. [30] or Tromas et al. [31] can be applied to the nanoindentation tests to cluster the acquired data into different domains. The only difference between our treatment and the one proposed by Ulm et al. is the objective function. Our method uses the absolute difference normalized by the maximum, whereas Ulm’s method takes the difference between the sum of the Gaussian cumulative density function and the experimental cumulative density function. This makes it possible to give the same weight to each experimental data point. In both cases, the minimization is performed simultaneously on the elastic modulus and the hardness.

## 3. Results

### 3.1. Macroscopic Structure

Figure 2 shows the structure across the thickness of an as-cast ingot obtained by brittle fracture. One can notice that the grain shape and size repartition is highly heterogeneous. The grains are smaller in the lower part, which was in contact with the copper mold, and more elongated and extended in the upper part, which was close to the free surface. It is possible to identify small intergranular cracks (from one-tenth of a millimeter to several millimeters), either horizontal or parallel to the free surface. These are probably sinking due to the inability of the solidifying material to accommodate deformation during cooling of the ingot.

### 3.2. Ultrasonic Elastic Characterization Results

Ultrasound measurements were carried out on the different samples. After measurements, the mean velocity of elastic waves was 7100 m/s with a standard deviation of 360 m/s. From the velocities measured, the E/A  ratio was evaluated to be 354 ± 15 GPa. Assuming a fixed Poisson’s ratio of 0.3, the calculated elastic modulus is 262 ± 10 GPa.

### 3.3. Compression Test Characterization Results

Seven samples were tested, but only three of them presented a real uniaxial and uniform compression field after measurement of the displacements by image correlation. Figure 3a shows the longitudinal, transverse and shear strains measured on these three samples. As expected, shear strains were nearly zero for the selected tests, confirming their uniaxial behavior. The transverse strains were positive, while the longitudinal ones were negative, confirming that compressions were involved. The overall tendency of the stress–strain curve shows that Ti_0.5_Fe_0.45_Mn_0.05_ has elasto-plastic behavior at the macroscopic scale. For all specimens, the stress–strain curve’s unloading and reloading phases are superimposed, confirming pure elastic behavior (Figure 3b). The elastic modulus was extracted from a linear regression over each unloading cycle.

Table 1 shows all the elastic moduli identified for the three samples with uniform compression loading. Each line contains between four and six identified elastic moduli, two for each completed unloading–reloading cycle. The last column gives the mean elastic modulus over all the cycles. The last two lines give, respectively, the mean elastic modulus over all the samples for each step and the mean elastic modulus over all the tests. The overall mean corresponded to an elastic modulus of 198 GPa. However, this measure varied by 66 GPa between the minimum and maximum. Finally, it should be noted that the elastic modulus tended to increase with accumulated compression, from 186 GPa to 215 GPa.

### 3.4. Nanoindentation Measurements and Energy-Dispersive Spectroscopy Results

Figure 4 shows the two matrices as imaged by SEM (Figure 4a,b) compared with the indentation elastic modulus maps (Figure 4c,d) and the hardness maps (Figure 4e,f). The two matrices of 10 × 10 indents revealed a rather spread of raw estimates for both the elastic modulus and the hardness. The studied alloy, Ti_0.5_Fe_0.45_Mn_0.05_, exhibits strong heterogeneity at the microscale. One can observe very tight correlations between the imaged microstructure, local elastic modulus and local hardness. Plotting the indentation modulus as a function of the hardness revealed four distinct domains (Figure 5). This was confirmed by the clustering analysis of the 200 concatenated values of indentation elastic moduli and hardness as illustrated in Figure 6.

Spatially, as shown in Figure 4, these different domains are well correlated with microstructure. The analysis allows us to estimate the proportion of each domain. All numerical results and domain associations are summarized in the Table 2. By considering the participation of each domain to the material properties as the proportion found here, the mean indentation elastic modulus M for Ti_0.5_Fe_0.45_Mn_0.05_ was found to be 294 ± 60 GPa. The points of chemical analysis by EDS are numbered in Figure 4a and Figure 4b, respectively, on matrix 1 and matrix 2. For each matrix, analysis spots were chosen to test each domain several times. Table 3 contains the atomic percentage of each identified elements for the fifteen measurements. The Prob. Dom. column links the dosing point with the probable domain. For all dosing points, the main element was titanium, combined with iron and manganese in smaller proportion. Chromium, vanadium and aluminum were also identified in the alloy. These traces probably came from the impurities present in the presumably pure raw materials. Every point in domain IV showed an atomic percentage of more than 60% titanium, up to 85% titanium, although the expected atomic percentage is 50% for Ti_0.5_Fe_0.45_Mn_0.05_. Considering only Ti, Fe and Mn, two other groups emerged from Table 3. The first one, having an atomic percentage of titanium around 60% and encompassing points 1, 3 and 8 in matrix 1 and point 5 in matrix 2, correlates well with domain III. The second one, with an atomic percentage of titanium around 50% or slightly under (about 46%), corresponds to the expected formulation of the material and was found in the last two domains: I, homogeneous gray, and II, heterogeneous gray.

## 4. Discussion

The uniaxial compression tests revealed very energetic failures of the specimens, with sudden collapses and projections of multitudes of fragments. The failures occurred after two or three elastic unloading–reloading cycles, which nonetheless allowed us to estimate an average macro-compression elastic modulus of 198 GPa with a minimum of 181 GPa and a maximum of 220 GPa. Stiffening behavior with an increasing degree of compression was observed, with the average modulus of each unloading–reloading cycle increasing from 188.5 GPa to 196.5 GPa and then 215 GPa. This suggests that the pre-existing intergranular cracks visible in Figure 1 played a large role here. It is quite possible that they closed during compression but that their pre-existence and their spatial distribution create random brittleness lines between grains. This would induce great variability in the stress at failure. Unfortunately, this could not be studied in detail here because the DIC coating was damaged prior to complete failure, putting an end to the measurement of strains. Regardless, it would require a campaign with a very large number of samples and an ad hoc statistical analysis. The uniaxial compression solicitation is among the most difficult to obtain, in spite of the precautions taken for machine alignment and contact lubrication. This is the main reason for the choice of non-intrusive deformation measurements by digital image correlation. It served as an arbiter and allowed us to discard the tests as soon as the stress axis diverges. However, image correlation is designed to measure on flat surfaces, perpendicular to the optical axis, which is not the case here, but many high-quality studies have been conducted on tubes or cylinders [32]. By restricting the area of analysis for image correlation computation to the central 1/3 of the specimen, i.e., 3.3 mm width, the deviation in depth was kept to only 0.28 mm. This was optically tolerated because the lens had an adequate depth of field. The technique prevented us from estimating the Poisson’s ratio but did not degrade the accuracy of the longitudinal measurements and the macro-stiffness assessment. We have therefore determined an average macroscopic stiffness in compression, relevant enough to understand and model the behavior of particles subjected to compression during hydrogen-induced expansion in a constrained container. However, this assertion is subject to variability due to the initial damage to the ingot, i.e., variability not controlled during the casting process. As intermetallic storage alloys are, most of the time, crushed into a powder of millimetric particles before use, the cracks and grain distribution within the ingot should have a significant impact on the mechanical properties at the intra-particle scale.

Published studies about the elastic constants of TiFeMn-based alloys have not been found. However, it is possible to find those of TiFe [24] and TiFe_2_ (λ Laves phase) [33] in the literature. The values of the elastic constants of TiFe and TiFe_2_ were used to compute a three-dimensional representation of the Young’s modulus and Poisson’s ratio. Using the method of Vlassak and Nix [34], the indentation elastic modulus is also represented in Figure 7.

From the values computed in Figure 7, it is clear that the elastic responses of both single crystals from a uniaxial test should be anisotropic. For TiFe, the mean Young’s modulus was 271 ± 43 GPa, the minimum value being 213 GPa and the maximum 352 GPa. For Fe2Ti, the mean value was 252 ± 18 GPa (the extrema being 226 GPa and 295 GPa). Compared to these, the values encountered previously with compression tests appear to be very low, confirming the impact of pre-existing fissures on the response of the material. Concerning the Poisson’s ratio, the mean value was 0.26 ± 0.04, the minimum being 0.19 and the maximum 0.31, for TiFe. For TiFe_2_, the mean value was 0.28 ± 0.02 (the values lying between 0.22 and 0.3). Under indentation, this anisotropy was greatly reduced due to the multiaxiality of the mechanical solicitation. The mean indentation elastic modulus for FeTi is 284 ± 6 GPa, the minimum value being 274 GPa and the maximum 295 GPa. For TiFe_2_, the mean indentation modulus was lower, 274 ± 5 GPa, the values lying between 266 GPa and 282 GPa.

Assuming isotropic and homogeneous macroscopic elastic behavior and combining the results obtained by ultrasonic and nanoindentation characterizations, it is possible to deconvolve the Young’s modulus from the Poisson’s ratio. As mentioned previously, both experimental responses give the following:(7)$$\frac{E}{1-\nu^{2}}=294\pm 60\;GPa,$$
(8)$$\frac{E\left(1-\nu \right)}{\left(1-2\nu \right)\left(1+\nu \right)}=354\pm 15\;GPa,$$

Combining these two equations yields
E=269±60 GPa,
(9)ν=0.29±0.03.

These values are in accordance with the ones mentioned previously. However, the standard deviation observed for nanoindentation, ±60 GPa, is not in accordance with the one predicted by Vlassak and Nix’s model, ±6 GPa, at least for the cubic phase of TiFe. This deviation is more comparable to the ones observed for each phase identified in nanoindentation.

This is why an analysis by nanoindentation and a comparison with microscopy and chemical analysis (EDS) were of prior interest. As presented in the results (Section 3.4), by clustering the indentation results, it was possible to identify four mechanical behaviors by their similarities in hardness and elasticity. These four domains were also visually identified on the SEM images (Figure 4). Based on the EDS measurements presented in Table 3, the heterogeneity of the elastic behavior at a small scale is discussed here. The stiffest domain (area 4: M = 384 ± 32 GPa) corresponded to dark spots on the SEM micrograph. It consisted of irregularly distributed spherical inclusions with diameters smaller than 5 µm. Their small size made it difficult to measure reliably both their chemical composition and their rigidity. The rest of the sample was composed of three phases whose compositions were closer to the expected Ti_0.5_Fe_0.45_Mn_0.05_. They could be discriminated thanks to nanoindentation or observed on the SEM micrographs. The estimated local indentation elastic moduli were between 265 GPa and 333 GPa.

From Table 3, it is possible to extract an average composition for domains 1, 2 and 3. Specifically, domain 1 had a composition of Ti_0.5_Fe_0.4_Mn_0.1_, domain 2 Ti_0.45_Fe_0.39_Mn_0.16_ and domain 3 Ti_0.6_Fe_0.3_Mn_0.1_. Domain 4 appeared to be very poor in Mn and seemed to exhibit carbon or nitrogen in large proportions. Accordingly, the formation of nitride or carbide precipitate can explain the very high stiffness observed for domain 4. Based on the liquidus projection of the ternary phase diagram of Ti-Fe-Mn shown in Figure 8 (from Figure 3, reproduced from Materials Science International Team (MSIT^®^) et al., 2008 [35]), it is clear that the cooling of the alloys with the composition of domains 1 and 2 involves λ Laves phases, while the cooling of the alloy with the composition of domain 3 does not. In this way, the precipitation of alloys I and II will lead to a mixture of λ and cubic TiFe-type phases, while the precipitation of alloy type 3 will lead to cubic TiFe-type phases. As the theoretical indentation modulus of pure TiFe_2_ (λ Laves phase) is slightly lower than that of TiFe, the hierarchy of elasticity is respected between domain I/II and domain III. However, the difference between domain I/II and domain III is experimentally almost 50 GPa, while the difference predicted between TiFe and TiFe_2_ by indentation is only 10 GPa. A first interpretation is that the addition of Mn to this phase may significantly widen this gap.

## 5. Conclusions

To the best of our knowledge, there is currently little or no experimental or theoretical data on the mechanical properties of Ti_0.5_Fe_0.45_Mn_0.05_ or, more broadly, of TiFe-based storage alloys. This paper is one of the first to give elasticity values for specific phases of a TiFeMn alloy at different scales. The following conclusions can be drawn:Through compression testing, it was possible to assess to the elastic modulus representative of the macroscopic behavior of the bulk material, considering all its defects (pre-existing cracks) and its microstructural heterogeneity. The encountered value, between 163 and 229 GPa, seems to increase with deformation due to closure cracks. In this way, the values encountered on compression appear slightly low compared to those expected for a homogenous, isotropic and perfect material.Combining ultrasonic wave propagation and instrumented indentation, it was possible to extract a representative Young’s modulus and Poisson’s ratio for each of the different phase present in the material. In fact, both techniques allowed us to obtain a numerical value of the modulus as a function of the Poisson’s ratio. A Young’s modulus of 269 GPa and a Poisson’s ratio of 0.29 were found. These values are in agreement with the one encountered in the literature and determined by an ab initio technique, at least for TiFe and TiFe_2_, but with a high standard deviation due to phase heterogeneity among the different phases present in the material.A closer examination of the local indentation response together with microstructural and chemical analysis allowed us to attribute specific elastic behavior to each identified phase present in the material, four in total. In particular, depending on the local proportions of Fe, Ti and Mn, the indentation elastic modulus of the TiFeMn alloy may vary from 265 to 333 GPa and the hardness from 9.2 to 18.7 GPa. Additionally, the presence of nitride or carbide phase, having an indentation elastic modulus of 385 GPa and a hardness of 25.4 GPa, has also been evidenced.

## Figures and Tables

**Figure 1 materials-17-06100-f001:**
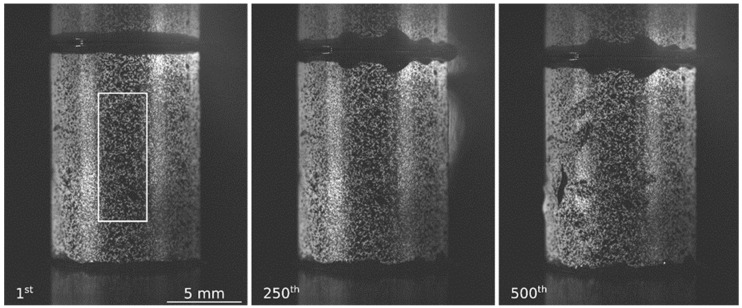
Raw images of Ti_0.5_Fe_0.45_Mn_0.05_ sample subjected to compression. First image prior to complete contact establishment; 250th image (282 MPa stress and a longitudinal strain of *ε* = 1.33%) and 500th image showing damage and coating peeling (668 MPa stress, unavailable strain).

**Figure 2 materials-17-06100-f002:**
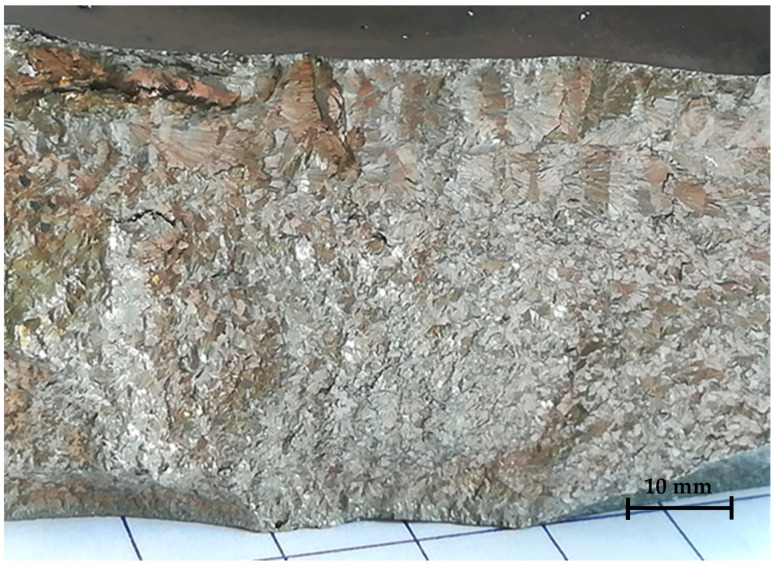
Surface generated by brittle fracture of a 40 mm thick rectangular Ti_0.5_Fe_0.45_Mn_0.05_ ingot. The lower surface was in contact with the copper mold, while the upper surface was a free surface.

**Figure 3 materials-17-06100-f003:**
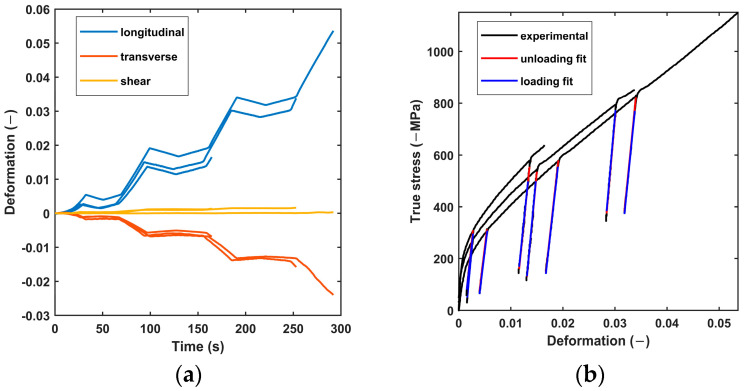
(**a**) Longitudinal, transverse and shear strain measured by DIC during compression tests on Ti_0.5_Fe_0.45_Mn_0.05_ and (**b**) the corresponding true stress–strain curve.

**Figure 4 materials-17-06100-f004:**
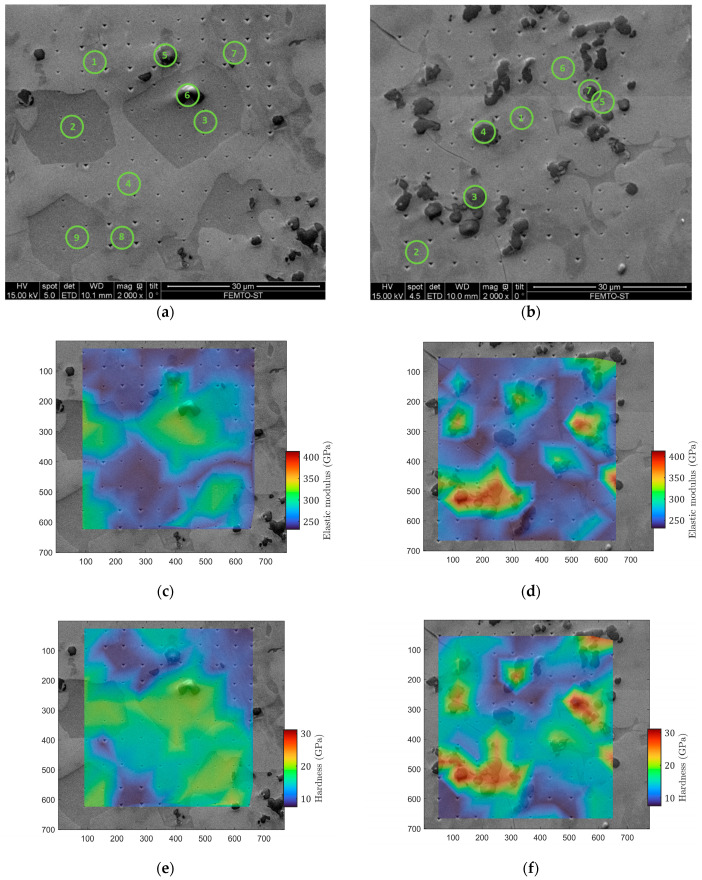
Comparison between SEM images (**a**,**b**) and associated indentation elastic modulus maps (**c**,**d**) and hardness maps (**e**,**f**). On images (a,b), numbers refer to EDS dosing point (see Table 3).

**Figure 5 materials-17-06100-f005:**
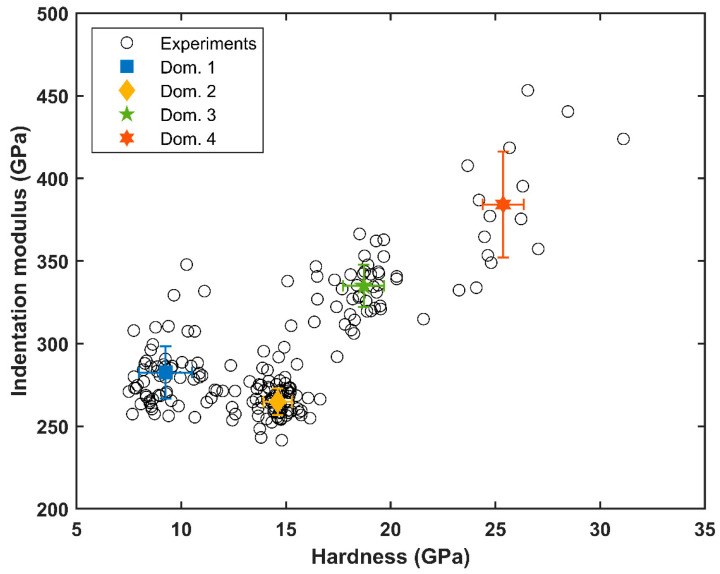
Indentation elastic modulus vs. hardness obtained for the 200 indentation measurements.

**Figure 6 materials-17-06100-f006:**
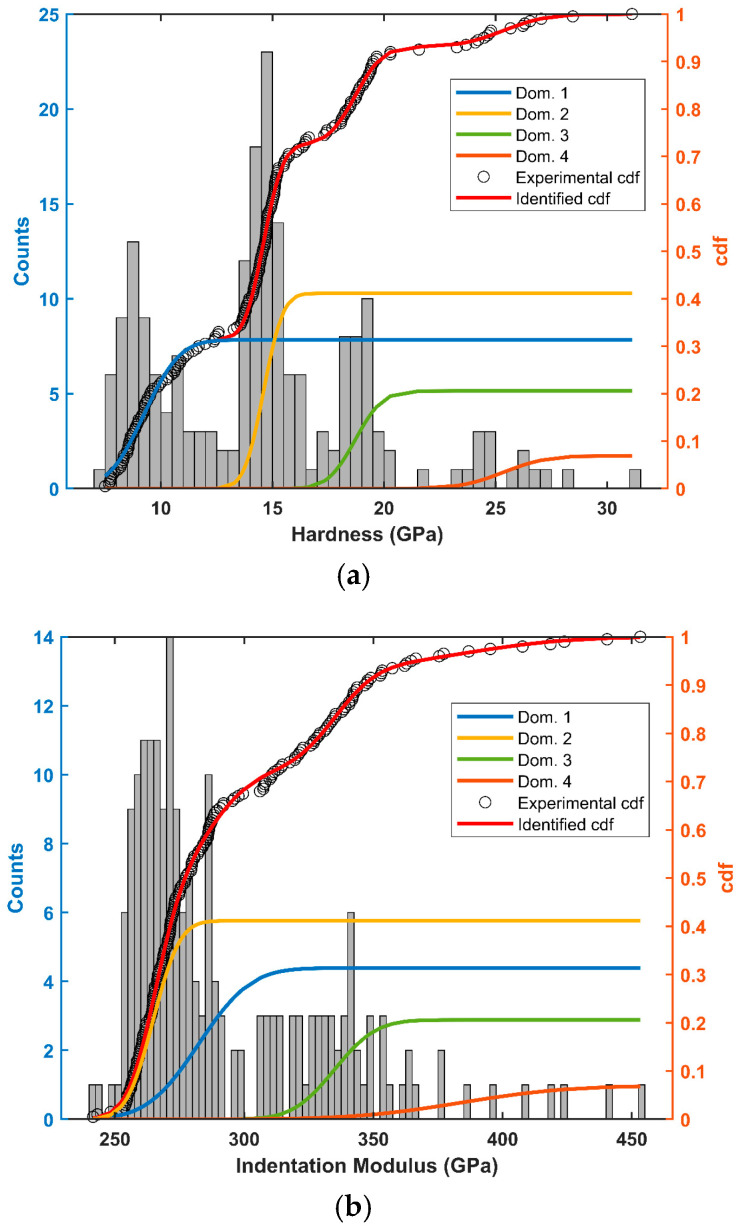
Histograms and cumulative density functions of (**a**) hardness (GPa) and (**b**) estimated indentation elastic modulus (GPa).

**Figure 7 materials-17-06100-f007:**
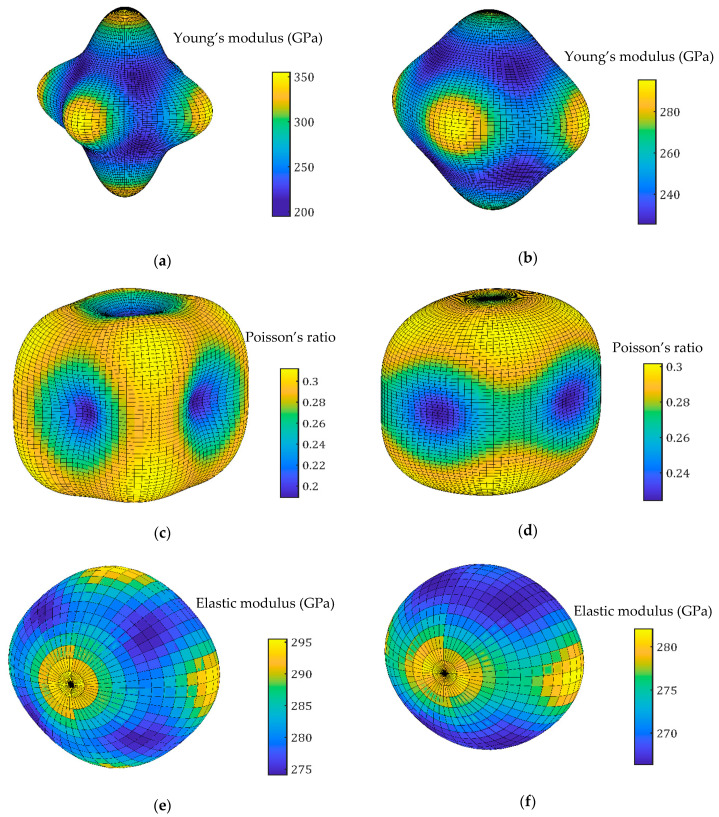
Three-dimensional representations of the Young’s modulus, Poisson’s ratio and indentation elastic modulus, respectively, for TiFe (**a**,**c**,**e**) and TiFe_2_ (**b**,**d**,**f**).

**Figure 8 materials-17-06100-f008:**
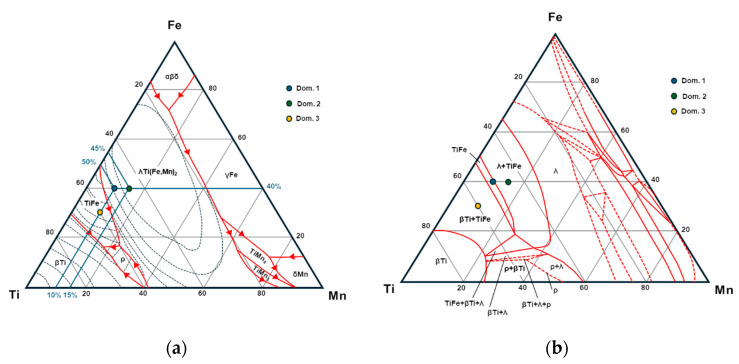
Ternary phase diagram for TiFeMn adapted with permission from Ref. [35] 2024: (**a**) liquidus surface projection and (**b**) isothermal section at 1000 °C.

**Table 1 materials-17-06100-t001:** Experimental elastic moduli, in GPa, measured during compression tests in the different unloading and reloading steps.

	First Cycle		Second Cycle		Third Cycle		
Sample	Unloading	Loading	Unloading	Loading	Unloading	Loading	Mean
1	191	177	197	194			189
2	216	217	213	216	229	229	220
6	166	163	180	178	200	200	181
Step mean E (GPa)	191	186	197	196	215	215	
Material mean E (GPa)							198

**Table 2 materials-17-06100-t002:** Domain identification of Ti_0.5_Fe_0.45_Mn_0.05_ based on data from over 200 nanoindentation tests.

Domain	Indentation Elastic Modulus (GPa)	Hardness (GPa)	Corresponding Color on SEM Micrography	Proportion (%)
I	265 ± 8	14.6 ± 0.7	Tabby gray	41.2
II	282 ± 16	9.2 ± 1.3	Lightest gray	31.4
III	333 ± 13	18.7 ± 1	Darkest gray	20.8
IV	384 ± 32	25.4 ± 1.5	Black dots	7.3

**Table 3 materials-17-06100-t003:** Energy-dispersive spectroscopy dosing of a Ti_0.5_Fe_0.45_Mn_0.05_ sample.

Matrix	Dosing Point	Element (% at)									Prob. Domain
		Ti	Mn	Fe	Cr	V	Al	O	C	N	
1	5	61	2	5					32		IV
	1	60	10	31							I
	8	58	10	32							III
	3	57	9	32	<1						III
	6	50	9	35			<1	4			I
	2	50	8	28	<1			13			III
	7	50	10	36				5			I
	4	42	16	36		<1	5				II
2	4	85	4	10							IV
	3	71	3	7						19	IV
	7	69	3	10						18	IV
	5	67	8	25							III
	1	51	12	38							I
	2	50	8	38			4				I
	6	43	15	36		<1	5				II

## Data Availability

The original contributions presented in the study are included in the article, further inquiries can be directed to the corresponding author.

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
