# Peer review of "Multiscale Elastic Modulus Characterization of Ti0.5Fe0.45Mn0.05, an Iron–Titanium–Manganese Alloy Dedicated to Hydrogen Storage"

_materials, 2024, doi:10.3390/ma17246100_

Round 1
Reviewer 1 Report
Comments and Suggestions for Authors
Hydrogen storage is an important matter in question of the directed investigations. Ti0.5Fe0.45Mn0.05 alloy was chosen as the study material due to its a good intermetallic storage compound.
Authors of the paper chose direction of the studies directed to elastic modulus characterization. The paper is written using vivid language, very well chosen material for the studies along with its justification (including addition of a small amount of Mn). Swelling and shrinking of the bed during the hydrogen absorption cycles was the centre for the studies. A violent change in volume at the crystal scale occurs under that reversible hydriding process.
The idea and the subject of the studies are well displayed so that the novelty of the paper is clear. Experimental part, method and results of the study along with discussion provided foundation to present some useful final conclusions. The critical side of the study may be the next step, if the followers are going to choose the path to invetigate next compounds and/or alloys.
Text of the manuscript needs attention to remove faults and mistakes. The PDF file with color highlights is enclosed as a reference.

Author Response
Dear,
All suggestions have been reported in the corrected version.
The reason for which mentions to Figure or Table are highlighted is not clear to us, so no changes were brought about it.
Best regards

Reviewer 2 Report
Comments and Suggestions for Authors
The paper is very well written and contributes experimental data on Multiscale elastic modulus characterization of Ti0.5Fe0.45Mn0.05 alloys. However, there are some problems, which must be solved before it is considered for publication. If the following problems are well-addressed, this reviewer believes that the essential contribution of this paper are important for the engineering applications of hydrogen storage alloys.
1 Abstract: Ultrasonic elastic characterization is not reflected in the abstract, modulus of elasticity is a very important engineering parameter, it is recommended to add the significance of different scales of characterization in the abstract
2 line 68 TiFeMn alloys have been studied, how was the ratio of Ti0.5Fe0.45Mn0.05 designed in this study?
3 Experimental part: A detailed description of the parameters of the sample preparation is recommended.
line 105 Suggests adding information about manufacturers of raw materials.
line118-119 Suggested Scale Dimensions be added to Figure 1.
4 line 248 What do the different samples mean? What is the difference? How is this different from the 7 samples in line 253?
5 Are the EDS test points in Figure 4 a correspondence to Table 3? In addition, I don't quite understand the difference between the test materials in Fig. 4a and Fig. 4b. It is suggested to add the particle size information of the raw materials, because there is a difference in the uniformity of Mn distribution between the two materials in Table 3.
It is recommended that a ruler and icon be added to Figure 4.
6 Overall, there are relatively few grammatical errors in the article, but there are still some areas that can be improved to enhance the accuracy and readability of the article.
Author Response
Comment 1: Abstract: Ultrasonic elastic characterization is not reflected in the abstract, modulus of elasticity is a very important engineering parameter, it is recommended to add the significance of different scales of characterization in the abstract
Response 1: Ultrasonic is present in the abstract (l.17).
Comment 2: line 68 TiFeMn alloys have been studied, how was the ratio of Ti0.5Fe0.45Mn0.05 designed in this study?
Response 2: This study deals with the mechanical characterization of TiFe-based alloy. Full justification of the studied composition is proposed (in corrected version) from line 68 to 77.
Comment 3: Experimental part: A detailed description of the parameters of the sample preparation is recommended.
line 105 Suggests adding information about manufacturers of raw materials.
line118-119 Suggested Scale Dimensions be added to Figure 1.
Response 3: The details of the preparation of the samples have been completed (name of ingot supplier has been added):
“The ingot is produced by MAHYTEC company using an induction furnace at 1300°C, under controlled atmosphere (argon), by melting the supposedly pure raw components (purity superior than 99.3%) into a zirconium dioxide crucible to be cast in a rectangular copper mold. The alloy is then cooled in the furnace still under argon gas.”
Scale in Figure 1 has been added.
Comment 4: line 248 What do the different samples mean? What is the difference? How is this different from the 7 samples in line 253?
Response 4: There is a misunderstanding. There are no two samples, but a single one with two indented zone. This has been added in the text. Once again it illustrates the heterogeneity of the measured mechanical properties locally.
Comment 5: Are the EDS test points in Figure 4 a correspondence to Table 3? In addition, I don't quite understand the difference between the test materials in Fig. 4a and Fig. 4b. It is suggested to add the particle size information of the raw materials, because there is a difference in the uniformity of Mn distribution between the two materials in Table 3.
It is recommended that a ruler and icon be added to Figure 4
Response 5: It is the same material but different areas are investigated. There’s a scale (ruler) on Figure 4 a) and 4 b). That’s the point, there’s a correspondence between Table 3 and Figure 4. It is what we intend to discuss.
Comment 6: Overall, there are relatively few grammatical errors in the article, but there are still some areas that can be improved to enhance the accuracy and readability of the article.
Response 6: Thank you for comment. We had a careful reading of our paper to identify some grammatical errors and to solve them.
Best regards

Reviewer 3 Report
Comments and Suggestions for Authors
Comments and Suggestions for Authors
In this paper, the authors provide a study about the mechanical characterization of Ti0.5Fe0.45Mn0.05 alloy at different scales. First, the elastic modulus is measured by compression tests over cylindrical samples. The estimated macroscopic elastic modulus is about 198GPa with a large variability, from 163 to 229GPa. Secondly, ultrasonic elastic characterization together with instrumented indentation allowed an estimation of both Young’s modulus and Poisson’s ratio, respectively at 269GPa and 0.29. Finally, the nano-indentation results confronted to SEM imaging and EDS analyses, reveal that several metallurgical phases co-exist under the particle scale.
The document is interesting and well structured. Some comments and suggestions for authors could be considered.
1. Move the following paragraph to 1. Introduction section. “The studied material is a Titanium Iron alloy with substitution of iron by manganese: Ti0.5Fe0.45Mn0.05. The TiFe alloy is the main representative of the AB family of solid hydrogen storage material. The physical and chemical properties have been widely studied and mechanical properties of TiFe have been derived from the crystalline structure by Benyelloul et al. [24]”.
2. Why is showed the surface generated by brittle fracture of a 40mm thick rectangular Ti0.5Fe0.45Mn0.05 in material and methods?
3. Move the following paragraph and Figure 1 to 3. Results section. “The Figure 1 shows the microstructure across thickness of a as cast ingot obtained by brittle fracture. One can notice that the grain shape and size repartition is highly heterogeneous. They are tinier in the lower part, in contact with the copper mold, and more elongated and extended in the upper part close to the free surface. It is possible to identify intergranular small cracks (from a tenth to several millimeters), either horizontal or parallel to the free surface. These are probably sinking due to the inability of the solidifying material to accommodate deformation during cooling of the ingot”.
4. Indicate the manufacturer's name, city and country of origin of all experimental equipment used in the study.
5. Use superscript of mm2 in lines 199 and 205.
6. Move the following paragraph and equations (5) and (6) to 2.2.2. Compression test section. “Assuming a volume conservation during the deformation, engineer strains, 𝜀0, are deduced from longitudinal true strains, 𝜀, and then true stress, 𝜎, are calculated from the following formulae:”
7. Figure 3 a) longitudinal, transversal and shear strain must be described in the text of the document.
8. In Figure 4, the identification of (a) to (f) is missing.
9. In Figure 4 (c) and (d) provide the units over the scale of “elastic modulus (GPa)”.
10. In Figure 4 (e) and (f) provide the units over the scale of “Hardness (GPa)”.
11. In Figure 7, the identification of (a) to (f) is missing.
12. In Figure 7 provide the units over the corresponding scales.
13. In conclusions 1 and 2, provide quantitative results of elastic modulus for compression, ultrasonic waves propagation and instrumented indentation analyzing the differences of results among the methods.
Author Response
Dear,
All changes are yellow highligted in the new version.
Comment 1: Move the following paragraph to 1. Introduction section. “The studied material is a Titanium Iron alloy with substitution of iron by manganese: Ti0.5Fe0.45Mn0.05. The TiFe alloy is the main representative of the AB family of solid hydrogen storage material. The physical and chemical properties have been widely studied and mechanical properties of TiFe have been derived from the crystalline structure by Benyelloul et al. [24]”.
Response 1: The paragraph has been moved (l68 to 72).
Comment 2&3: Why is showed the surface generated by brittle fracture of a 40mm thick rectangular Ti0.5Fe0.45Mn0.05 in material and methods?
Move the following paragraph and Figure 1 to 3. Results section. “The Figure 1 shows the structure across thickness of a as cast ingot obtained by brittle fracture. One can notice that the grain shape and size repartition is highly heterogeneous. They are tinier in the lower part, in contact with the copper mold, and more elongated and extended in the upper part close to the free surface. It is possible to identify intergranular small cracks (from a tenth to several millimeters), either horizontal or parallel to the free surface. These are probably sinking due to the inability of the solidifying material to accommodate deformation during cooling of the ingot”.
Response 2&3: It has been moved to Results section. 3.1 Macroscopic structure (l242 to 252)
Comment 4: Indicate the manufacturer's name, city and country of origin of all experimental equipment used in the study.
Response 4: We do not understand the purpose of this request. The names of equipment (and so the manufacturer) are furnished and conditions of use are established. If further details are required, we will respond if a more specific demand is made.
Comment 5: Use superscript of mm2 in lines 199 and 205.
Response 5: The corrections have been made.
Comment 6: Move the following paragraph and equations (5) and (6) to 2.2.2. Compression test section. “Assuming a volume conservation during the deformation, engineer strains, ?0, are deduced from longitudinal true strains, ?, and then true stress, ?, are calculated from the following formulae:”
Response 6: The paragraph has been moved (l198 to 202).
Comment 7: Figure 3 a) longitudinal, transversal and shear strain must be described in the text of the document.
Response 7: The following paragraph has been added (l265-267):” As expected, shear strains are nearly zero for the selected tests, confirming their uniaxial behaviors. The transversal strains are positive while the longitudinal ones are negative confirming that compressions are involved.” To comment the shape of the strain curves.
Comment 8: In Figure 4, the identification of (a) to (f) is missing.
Response 8: Done
Comment 9&10: In Figure 4 (c) and (d) provide the units over the scale of “elastic modulus (GPa)”. In Figure 4 (e) and (f) provide the units over the scale of “Hardness (GPa)”.
Response 9: Done
Comment 11: In Figure 7, the identification of (a) to (f) is missing.
Response 11: Done, in Figure 6 also added
Comment 12: In Figure 7 provide the units over the corresponding scales.
Response 12: Done
Comment 13: In conclusions 1 and 2, provide quantitative results of elastic modulus for compression, ultrasonic waves propagation and instrumented indentation analyzing the differences of results among the methods.
Response 13: The values have been recalled in points 1 and 2 (l455 and l452)
Regards
